# Spatio-Temporal Pattern of Urban Green Space in Chengdu Urban Center under Rapid Urbanization: From the Policy-Oriented Perspective

Kelei Li [1], Wenpeng Du [1,2,*], Zhiqi Yang [3], Huimin Yan [2,4] and Yutong Mu [5]

1   Institute of Geography and Resources Science, Sichuan Normal University, Chengdu 610101, China; 2021100623@stu.sicnu.edu.cn
2   Institute of Geographic Sciences and Natural Resources Research, Chinese Academy of Sciences, Beijing 100101, China; yanhm@igsnrr.ac.cn
3   School of Political Science and Public Administration, Soochow University, Suzhou 215031, China; zqyang@suda.edu.cn
4   University of Chinese Academy of Sciences, Beijing 100049, China
5   Xi'an Yaozhizhongchuang Land Survey and Planning Co., Ltd., Xi'an 710075, China; 2015227006@chd.edu.cn
*   Correspondence: duwp@sicnu.edu.cn

**Abstract:** Urban green space (UGS) is increasingly recognized as a nature-based solution to achieving urban sustainable development. Under rapid urbanization, greening policies are often the main driving factor behind the restoration or even growth of UGS. In this study, Chengdu, the pioneering "park city" in China, is chosen as a representative region. Based on land use/land cover (LULC) and normalized difference vegetation index (NDVI) data, indicators that can reflect both UGS quantity and quality are constructed and the spatio-temporal characteristics of UGS in original and expanding urban areas are also explored at different greening policy stages. The findings show that, from 2000 to 2022, the basic trend of UGS reduction during urbanization remained unchanged, despite the greening policies implemented in Chengdu. However, the original urban area has evolved into a new urban area. This has been achieved by integrating the expanded urban area with higher greening rates, resulting in the greening rate in 2022 (44.61%) being restored to the 2000 level (44.21%). The implementation of green policies in Chengdu is primarily reflected in improved UGS quality, especially in the stage of the ecological garden city construction (2008–2018). Specifically, the UGS quality in the original urban area has been improved by 25.25%. Overall, the UGS quality in Chengdu Urban Center has improved, changing from a medium level in 2000 to a medium-high level in 2022. The construction of a national demonstration zone of the park city provides an opportunity for the UGS quantity to increase and quality to improve in Chengdu in the future. However, effectively considering the development positioning of the Tianfu Granary to coordinate the relationship between UGS and high-quality farmland is a huge challenge for urban sustainable development in Chengdu.

**Keywords:** urban green space; quantity–quality; urban center; spatial-temporal pattern; greening policy; Chengdu

## 1. Introduction

Over the past 50 years, the world has witnessed a rapid increase in the urbanization process, with a doubling of the urban population and an increase of over 150% in built-up areas [1,2]. Significant disparity currently exists in the urbanization progress among countries worldwide [3]. For developing nations, challenges such as the urban heat island effect, environmental pollution, and biodiversity loss due to haphazard urban expansion have emerged as crucial impediments to achieving sustainable development [4–7]. Consequently, urban sustainable development has garnered considerable attention and become one of the Sustainable Development Goals (SDGs) [8,9].

Urban green space (UGS) is an artificial, semi-natural, and natural ecosystem in urban areas dominated by vegetation, including parks, gardens, forests, grasslands, and nature reserves [10,11]. It can provide a multitude of valuable ecosystem services and play a pivotal role in mitigating the urban heat island effect, reducing the urban flooding risk, and enhancing the well-being of urban residents [12–14]. Therefore, UGS construction has become a nature-based solution to promoting urban sustainable development [15,16]. In terms of the research trajectory, prior to the 21st century, studies on UGS primarily focused on their significance in relation to the urban environment. The narrow scope of the study was due to limitations in data availability and accuracy [17,18]. In the 21st century, with the development of remote sensing technology, UGS studies have gradually shifted away from exploring its spatio-temporal evolution characteristics [19,20] towards focusing on its accessibility, inequality, and coupling relationships with human well-being [14,21–23]. Although the hotspots in related fields are constantly changing, recognizing the spatio-temporal characteristics of UGS is the basis for conducting subsequent series of studies.

As previously mentioned, the development of remote sensing technology has significantly contributed to the extensive and in-depth investigation of UGS [24]. Land use/land cover (LULC) and vegetation indices (VIs) are widely employed as remote sensing indicators for mapping UGS [18,25]. Labib et al. [26] conducted an analysis of 93 relevant literatures, revealing that over 70% of the findings were based on these aforementioned indicators for studying UGS. Additionally, several specialized indicators for UGS research, such as the urban neighborhood green index (UNGI) and urbanization-vegetation cover coordination index (UVCI), have also been developed based on LULC and VIs [27,28]. In comparison to LULC, VIs can effectively capture higher-quality information pertaining to UGS [29,30]. Therefore, VIs are frequently utilized in exploring the interconnected relationship between UGS and human well-being [14,31].

During the 21st century, China has experienced the fastest urban population growth and the largest expansion of urban areas in the world [3]. Under the backdrop of rapid urbanization, greening policies have emerged as a crucial driving force for UGS recovery and growth [32,33]. Due to the fact that quantifying greening policy factors is difficult, the vast majority of studies have only qualitatively discussed the correlation between UGS and greening policies [34,35]. Quantitative studies have been further limited to indirectly showing that UGS is influenced by greening policies by analyzing how UGS changes are not solely influenced by urbanization, economic development, and climate [36]. Currently, UGS research is clustered around the insights that it brings to relevant policy making [37,38]. Urbanization has led to the evolution of greening policies; however, the two are not yet fully synchronized [39]. Therefore, while some attention has been paid to exploring the relationship between UGS and the associated greening policies in different urbanization stages [20,40], a gap still remains in terms of understanding how different stages of greening policy implementation impact changes in UGS. Furthermore, previous studies have examined spatial changes in UGS between original and expanding urban areas at various stages of urbanization [28]. However, little attention has been paid to differences in spatial changes between these areas at different stages of greening policy implementation. Given the variations in response speed to national greening policies and specific implementation programs across cities, addressing these issues at a municipal level is more meaningful.

Chengdu ranked 83rd in the comprehensive global city ranking[1] and is the core city of the Chengdu–Chongqing Urban Agglomeration, the largest urban agglomeration in Southwest China. According to official statistics, the urbanization rate of Chengdu was 79.89% in 2022, up 26.17% from 2000, and the built-up area of the central urban space was 1,064 km$^2$, about five times that of 2000[2]. Chengdu is known as the "National Forest City" and one of the "Most Happiness-Inspiring Cities"; this is also the place in China where the "Park City" was first proposed [41]. In addition, topics such as UGS fairness, UGS vitality, and UGS soil organic carbon mineralization have been explored in existing studies related to UGS in Chengdu [42–44]. However, the correlation between spatio-temporal changes in

UGS and greening policies has not yet been revealed. Therefore, selecting Chengdu as the case area to investigate the spatio-temporal pattern of UGS under the background of rapid urbanization and greening policies is both typical and representative.

This study therefore selects Chengdu as a representative study area and, firstly, divides the city into different greening stages by combining the greening policies implemented in the 21st century. Based on the LULC and VIs data, the spatio-temporal characteristics of UGS in Chengdu Urban Center are evaluated from the quantitative and qualitative dimensions, focusing on the differences between the spatial distribution of UGS at different greening stages, as well as the differences in the spatial changes of UGS between the original and the expanding urban areas. This study aims to reveal the spatio-temporal pattern of UGS in Chengdu under the dual background of rapid urbanization and greening policies, with aims of providing a reference for other cities to achieve urban sustainable development by the policy-guided construction of UGS.

The remainder of this paper is organized as follows: Section 2 describes the division of the greening policy stages in Chengdu, as well as the basic data and methods used in this study. Section 3 introduces the spatio-temporal changes of UGS in Chengdu, specifically focusing on the correlation between the changes with greening policies. Section 4 compares the UGS change in Chengdu with other cities, analyzes the opportunities and challenges of future UGS construction in Chengdu, and summarizes the limitations and future directions. Section 5 draws conclusions.

## 2. Materials and Methods

### 2.1. Division of the Stages of Greening Policies

From a careful analysis of policy documents related to greening policies since the 21st century, Chengdu's greening policies can be categorized into three stages based on the differences in urban development goals (Figure 1): the environmental protection model city construction stage (2000–2008), the ecological garden city construction stage (2008–2018), and the national park city construction stage (2018–2022). In the first stage, the goal was to create a beautiful environment and keep the streets clean and tidy. In the second stage, the goal was to significantly improve the eco-environment and enhance the function of urban–rural ecosystems. In the third stage, the goal was to integrate the parks and urban spaces and build a multi-tiered urban ecological greening system.

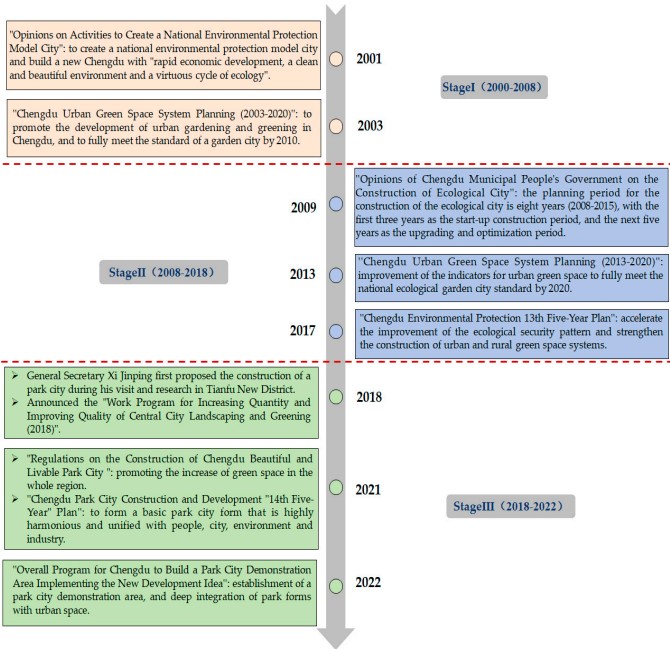

**Figure 1.** Time map for Chengdu's green policy promulgation.

### 2.2. Indicator Selection

Land use/land cover (LULC) and vegetation indices (VIs) are remote sensing indicators that can directly and accurately reflect quantitative and qualitative information of UGS, respectively [18,25]. Therefore, this study explored the spatio-temporal pattern of UGS in Chengdu based on LULC and VI data.

Compared to LULC, VIs are of various types; however, there are three types that are commonly used to reflect vegetated greenness, namely, the normalized difference vegetation index (NDVI), enhanced vegetation index (EVI), and near-infrared reflectance of vegetation ($NIR_v$) [45]. The NDVI is calculated based on near-infrared reflectance (841–876 nm) and red reflectance (620–670 nm), which is recognized as one of the most effective indicators for vegetation changes [46]. The EVI is calculated on the basis of near-infrared reflectance, red reflectance, and blue reflectance (400–460 nm), which can effectively reduce the atmospheric and soil noise effects and is often used to reveal vegetation changes in areas with high vegetation cover [47]. The $NIR_v$ is a recently developed vegetation index, calculated from NDVI and near-infrared reflectance, which can reflect vegetation phenology information; however, the response to climate elements is not clear [48,49].

Among the above, both the NDVI and EVI are frequently used in spatial investigations of UGS, with the NDVI being the most prevalent in UGS studies [26,28]. In comparison to the annual mean value of the NDVI ($NDVI_{ave}$), the annual maximum value of the NDVI ($NDVI_{max}$) exhibits greater sensitivity in reflecting vegetation information within urban areas [50]. Consequently, this study chooses the $NDVI_{max}$ as the basic index for the study of the spatio-temporal characteristics of UGS.

### 2.3. Data Source and Preprocessing

The basic data used in this study include land use data, urban boundary data, impervious cover data, and $NDVI_{max}$ data. The basic information is shown in Table 1.

**Table 1.** Basic data used in this study.

| Name | Resolution | Range Used in This Study | Data Sources | Notes |
|---|---|---|---|---|
| China land cover dataset, CLCD | 30 m | 2000, 2008, 2018, 2022 | Earth System Science Data | Evaluation of UGS quantity |
| Global urban boundary, GUB | 30 m | 2000, 2005, 2010, 2018 | Peng Cheng Laboratory | - |
| Global artificial impervious area, GAIA | 30 m | 2008, 2022 | Peng Cheng Laboratory | Access to GUB |
| NDVI | 30 m | 2000, 2008, 2018, 2022 | National Ecosystem Science Data Center | Evaluation of UGS quality |

Notes: The data download link is in section *Data Availability Statement*.

The urban boundary data are from the 30 m global urban boundary (GUB) dataset, which was produced by Prof. Peng Gong's team at Tsinghua University. The dataset is based on global artificial impervious area (GAIA) data and generated using the kernel density estimation approach and cellular-automat model. This dataset is now available for download in seven time phases (1990, 1995, 2000, 2005, 2010, 2015, and 2018) [52]. The GUB does not fully meet the needs of this study. Therefore, this study produced 2008 and 2022 urban boundary data on Chengdu based on GAIA data, referring to the algorithm in the "Mapping global urban boundaries from the global artificial impervious area (GAIA) data". As can be seen in Figure 2c, the urban boundary range of Chengdu in 2008 was between 2005 and 2010 and in 2022, the urban boundary range of Chengdu highly overlapped with the periphery of the built-up area.

The land use data are from the China land cover dataset (CLCD), produced by Prof. Yang and Huang of Wuhan University. This dataset reveals the land use and changes in China at a 30 m spatial resolution since 1990. The overall accuracy of the CLCD reached 79.31%, which outperforms the accuracy of MCD12Q1, ESACCI_LC, FROM_GLC, and GlobeLand30 [51]. Figure 2a shows the classification and land use in Chengdu in 2022.

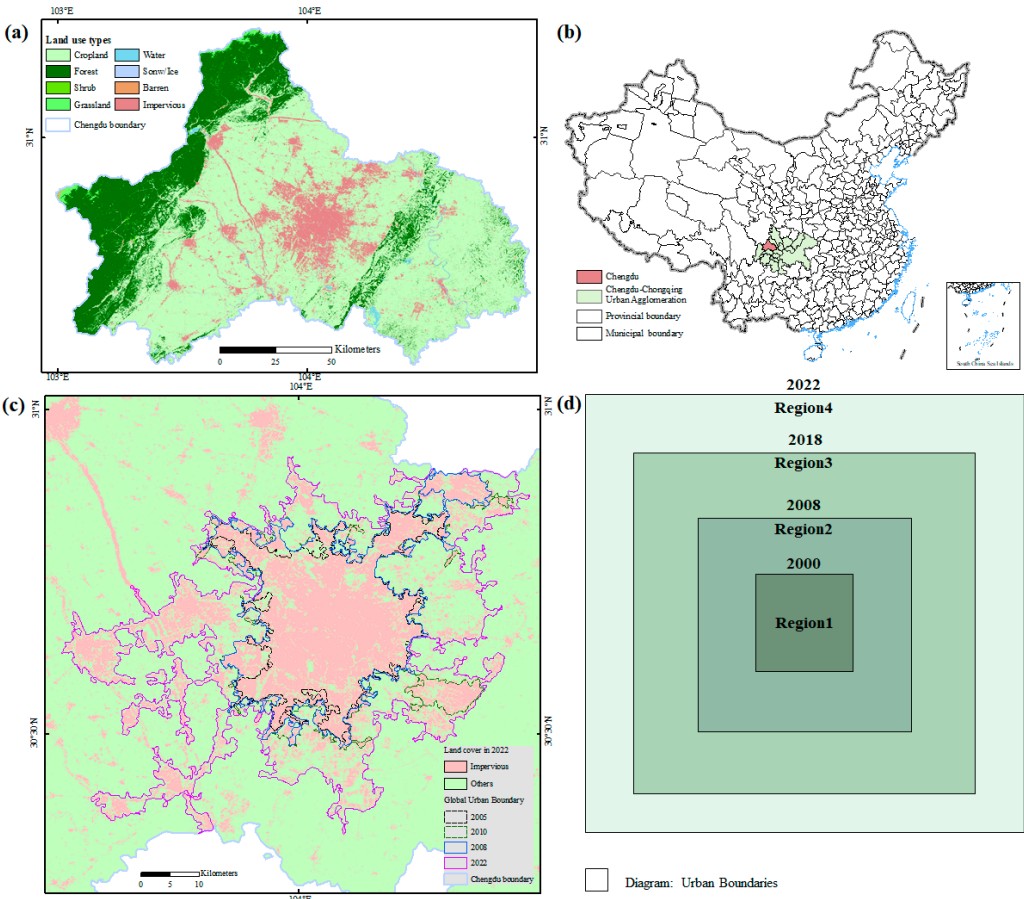

**Figure 2.** (**a**) Land use in Chengdu in 2022; (**b**) the localization of Chengdu in mainland China and the Chengdu–Chongqing Urban Agglomeration; (**c**) the spatial range of the urban boundary in Chengdu Urban Center; (**d**) schematic representation of the urban boundary range at different stages of greening policies.

The GAIA data used to produce the GUB originate from Prof. Peng Gong's team. The dataset provides yearly global impervious surface data (since 1985) and has been updated to 2022. The assessment of the GAIA accuracy in seven time phases (1985, 1990, 1995, 2000, 2005, 2010, and 2015) shows that the mean accuracy is higher than 90% [53].

The $NDVI_{max}$ data are from the team of Prof. Jingwei Dong at the Institute of Geographic Sciences and Natural Resources Research, Chinese Academy of Sciences. This dataset is based on the Google Earth Engine cloud computing platform. All Landsat5/7/8/9 remote sensing data throughout the year are used to obtain the maximum value of the NDVI in a year for each pixel, from 2000 to 2020. This is achieved by means of a series of data preprocessing and data smoothing actions [54].

### *2.4. Methods*
### 2.4.1. Definition of the Dynamic Urban Center Boundaries

With the rapid urbanization, the built-up area is expanding rapidly. In order to reflect the dynamic changes in the morphology of urban boundaries, this study defines different original and expanding urban areas for different stages of greening policy implementation in Chengdu (Figure 2d). During the environmental protection model city construction stage (2000–2008), the original urban area is Region 1 and the expanding urban area is Region 2. During the ecological garden city construction stage (2008–2018), the original urban area is Region 1+2 and the expanding urban area is Region 3. In the national park city construction stage (2018–2022), the original urban area is Region 1+2+3 and the expanding urban area is Region 4.

2.4.2. Quantitative Characteristics of UGS

According to the CLCD data, first, the spatial range of UGS is determined: five land use types, namely, cropland, forest, shrub, grassland, and wetland, are determined as UGS in this study. Then, the quantitative characteristics of regional UGS are reflected through the ratio of urban green space area (greening rate) within the region. The calculation formula is as follows:

$$QN = \frac{C_G}{C_R} \times 100\%$$

In the formula, $QN$ represents the greening rate (%), $C_R$ represents the total number of pixels in urban space, and $C_G$ represents the number of pixels corresponding to UGS within urban areas. In this study, UGS is determined by and based on CLCD data.

2.4.3. Qualitative Characteristics of UGS

The classification of UGS quality was conducted based on the $NDVI_{max}$ within UGS. The grading criteria are further divided into five categories, according to the research findings of Cai et al. [55] (Table 2):

**Table 2.** Urban green space quality grading scale.

| Quality Level | Grading Criteria | Description |
|:---:|:---:|:---:|
| 1 | $0.10 \leq \overline{NDVI_{max}} < 0.30$ | Low quality |
| 2 | $0.30 \leq \overline{NDVI_{max}} < 0.45$ | Medium-low quality |
| 3 | $0.45 \leq \overline{NDVI_{max}} < 0.60$ | Medium quality |
| 4 | $0.60 \leq \overline{NDVI_{max}} < 0.75$ | Medium-high quality |
| 5 | $0.75 \leq \overline{NDVI_{max}} < 1.00$ | High quality |

Note: $\overline{NDVI_{max}}$ indicates the average annual maximum NDVI value of UGS.

## 3. Results

### 3.1. Overall Change Trends of UGS in Chengdu

The greening rate of the urban center in Chengdu first decreased and has been increasing since the 21st century. In 2022, the greening rate was about 44.61% in the urban center, representing a recovery to the level of 2000 (44.21%). Encouragingly, the UGS quality of the urban center has experienced a significant increase, moving from 0.5092 in 2000 to 0.6280 in 2022 (an increase of approximately 23.33%), showing that the UGS transitioned from the level of medium quality to medium-high quality (Table 3).

From 2000 to 2022, there were significant differences in the quantity and quality of UGS changes in Chengdu's urban center at different stages of greening policy implementation. The rate of urban construction occupying UGS showed a decreasing trend. For example, in the original urban area, the greening rate average decreased by 2.79%, 1.65%, and 0.87% per year, respectively, over the three greening policy stages. Correspondingly, the UGS quality showed the characteristics of rapid deterioration, significant improvement, and slight improvement (Table 3).

In the environmental protection model city construction stage, the quantity and quality of UGS in the original and expanding urban areas experienced a significant decline. Notably, the decrease was more significant in the expanding urban area. In the ecological garden city construction stage, the UGS quantity in both the original and expanding urban areas still showed a significant decline. However, the rate of decline slowed, compared to the previous stage. The UGS quality showed an improvement and the UGS quality in the original urban area improved by 25.25%, which was significantly higher than that in the expanding urban area. In the national park city construction stage, the UGS quantity in both the original and expanding urban areas showed a slight decrease; whereas, the quality improved slightly (Table 3).

**Table 3.** Changes in the quantity (greening rate) and quality ($\overline{\text{NDVI}_{\text{max}}}$) of UGS in Chengdu Urban Center.

| | | Stage1: 2000–2008 | | Stage2: 2008–2018 | | Stage3: 2018–2022 | | 2022 |
|---|---|---|---|---|---|---|---|---|
| | | Original Urban Area | Expanding Urban Area | Original Urban Area | Expanding Urban Area | Original Urban Area | Expanding Urban Area | Urban Center |
| Area (km$^2$) | | 442.64 | 462.21 | 904.85 | 752.37 | 1657.22 | 576.42 | 2233.64 |
| Quantity | Start | 44.21% | 91.28% | 40.16% | 83.87% | 37.63% | 80.99% | |
| | End | 19.13% | 60.27% | 22.06% | 56.34% | 33.30% | 77.13% | 44.61% |
| | Change | −25.08% | −31.01% | −18.10% | −27.53% | −4.33% | −3.86% | |
| Quality | Start | 0.5092 | 0.6205 | 0.4629 | 0.5602 | 0.5842 | 0.6490 | |
| | End | 0.4401 | 0.4699 | 0.5798 | 0.5863 | 0.6086 | 0.6520 | 0.6280 |
| | Change | −13.57% | −24.27% | 25.25% | 4.66% | 4.18% | 0.46% | |

Notes: (1) In terms of quantity, the variation in the greening rate at the initial and final periods is used to characterize the change in the amount of UGS. In terms of quality, the amplitude of variation in the NDVI at the initial and final periods is used to characterize the change in UGS quality. (2) In this study, the spatial range of the urban center is dynamically changing in different greening stages. In the process of urbanization, the old central urban areas (original urban areas) and peri-urban areas (expanding urban areas) form new central urban areas. Usually, greening rates are significantly higher in expanding urban areas than in original urban areas. The green spaces in expanding urban areas also decrease during urbanization; however, this contributes to an increased greening rate in the new central urban areas, compared to the old ones. This also explains the decline in greening rates in both the original and expanding urban areas in all three stages in Table 3; however, the greening rate in the central urban area in 2022 is slightly higher than in 2000.

### 3.2. Spatial Pattern of UGS in Chengdu

Since the beginning of the 21st century, the implementation of greening policies in Chengdu has not changed the phenomenon of the UGS occupied during urbanization (Figure 3). The greening rate of both original and expanding urban areas in Chengdu has experienced a significant decrease, particularly during the environmental protection model city construction stage and ecological garden city construction stage (Figure 4a). Separately, in the environmental protection model city construction stage, the greening rate of the expanding urban area experienced a significantly greater decline, compared to that of the original urban areas in the eastern and southern regions, while both original and expanding urban areas in the western and northern regions exhibited a similar degree of decrease. In the ecological garden city construction stage, the greening rate in the expanding urban area experienced a significantly greater decline than that of the original urban areas in the eastern, southern, and western regions. Meanwhile, both original and expanding urban areas exhibited similar levels of decrease in the northern region (Figure 4a). Compared to the preceding two stages, the decline in the greening rate of the original and expanding urban areas in each region exhibited a significant deceleration during the national park city construction stage (Figure 4a).

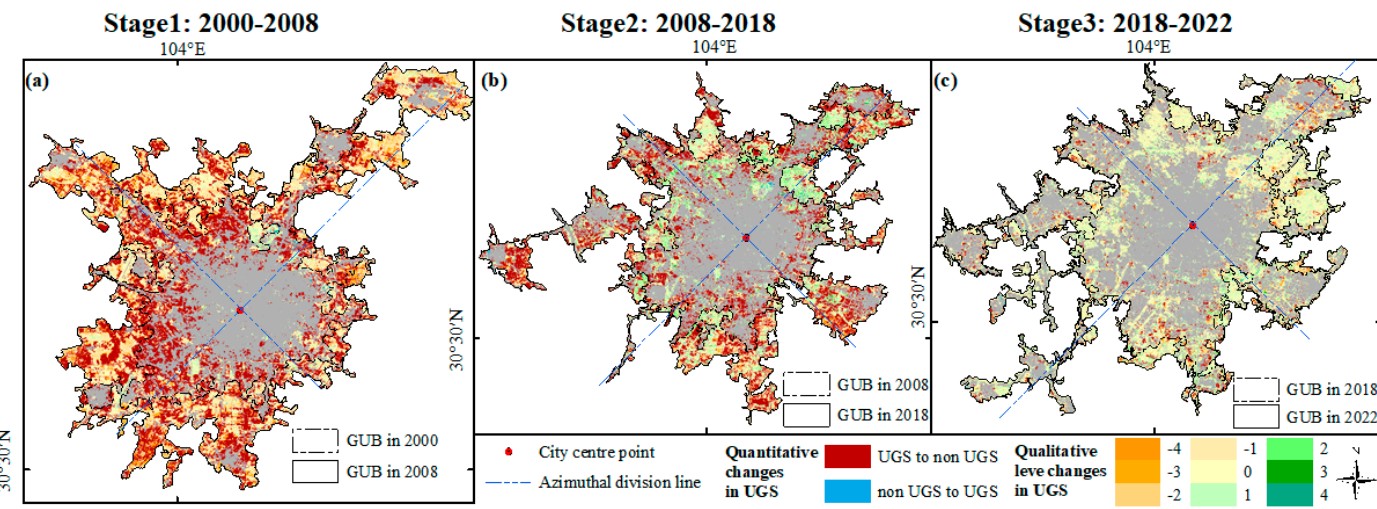

**Figure 3.** Spatial distribution of UGS quantity change and quality grade change in Chengdu Urban Center (with Tianfu Square, a landmark in Chengdu central city, dividing Chengdu into four parts: the eastern, southern, western, and northern regions).

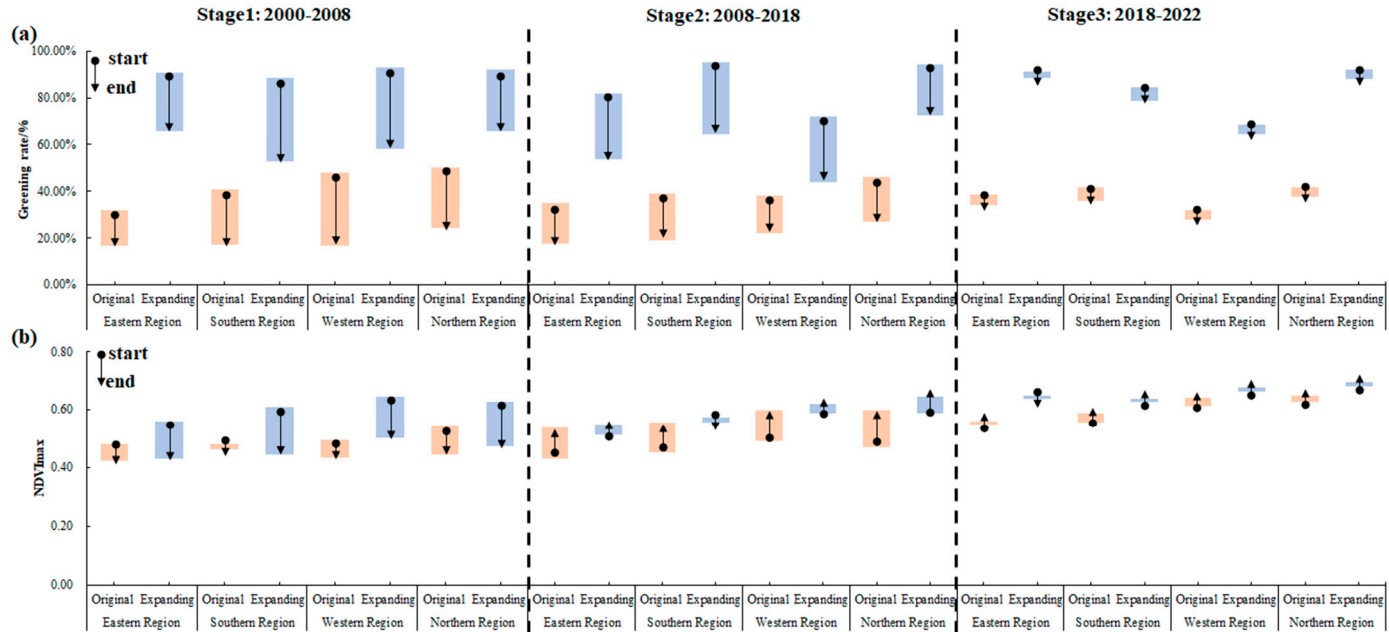

**Figure 4.** Changes in the quantity (greening rate) and quality ($\overline{\text{NDVI}_{\text{max}}}$) of UGS between original and expanding urban areas in different directions ((**a**): UGS quantity; (**b**): UGS quality).

In the environmental protection model city construction stage, the quality rating of UGS declined significantly more in other regions than it increased in Chengdu's urban center (Figure 3a). Specifically, both the original and expanding urban areas in the east, south, west, and north experienced decreases in UGS quality; the rate of decrease was also considerably higher in the expanding urban areas (Figure 4b). During the ecological garden city construction stage, apart from a slight decline in the UGS quality of incremental urban areas in the southern region, the UGS quality of both original and expanding urban areas in other regions showed improvement. Notably, there was a significant increase in the UGS quality within original urban areas while there was a relatively minor increase within expanding urban areas (Figures 3b and 4b).

During the national park city construction stage, with the exception of the eastern region (where there was a slight decrease in UGS quality in expanding urban areas), all

regions witnessed an increase in the UGS quality of both original and expanding urban areas. Although the increase in UGS quality was greater for original urban areas than for expanding ones, the disparity between them exhibited a significant reduction, compared to the preceding stages (Figures 3c and 4b).

During the urbanization, original urban areas integrated with expanding urban areas with relatively high greening rates to form new central urban areas. This resulted in the recovery or even growth of greening rates in the central urban areas of each region. The greening rate in the eastern region experienced a consistent increase, rising from 32% in 2000 to 52.65% in 2022, meaning the region had the highest greening rate in Chengdu (Figure 5a). The greening rate of the southern, western, and northern regions was the first to fall and then rise. Specifically, the southern region was the first to increase the greening rate during the ecological garden city construction stage. Specifically, the greening rate in 2022 (46.03%) exceeded the rate in 2000 (40.84%). However, the greening rates in the western and northern regions only increased during the national park city construction stage and neither greening rate had recovered to the 2000 level by 2022 (Figure 5a). The UGS quality in all regions of Chengdu first showed a decrease and then an increase and all regions started to improve from the ecological garden city construction stage (Figure 5b). By 2022, the UGS quality in the eastern and southern regions ($NDVI_{max} \approx 0.60$) was lower than that in the western and northern regions ($NDVI_{max} \approx 0.65$); however, all regions were at the medium-high quality level (Figure 5b).

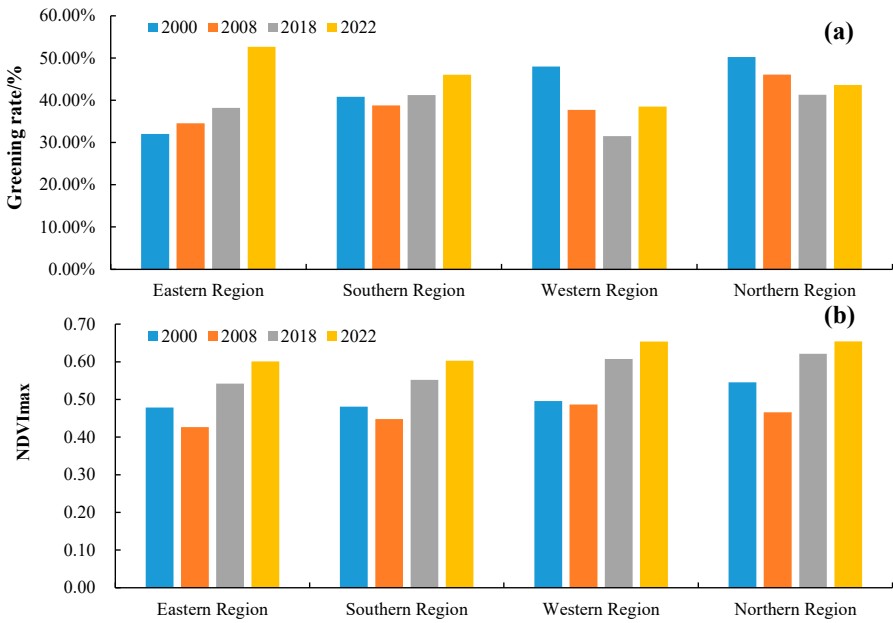

**Figure 5.** Changes in the quantity (greening rate) and quality ($\overline{NDVI_{max}}$) of UGS in the urban center of Chengdu in different directions ((**a**): UGS quantity; (**b**): UGS quality).

### 3.3. Relevance to Greening Policies

During the environmental protection model city construction stage, the main construction goal of Chengdu was to have a clean and beautiful environment (Figure 1). The requirement for the urban center at that time was only to keep the streets clean and tidy. In the document *Urban Green Space System Planning in Chengdu*, it was proposed that the eco-environment of the urban area should reach the standard of a garden city by 2010. However, according to the planning layout of green space parks in the document, the new or repaired green space parks were mainly to be located in suburban areas. The area of new greening facilities in built-up areas also dropped from 3.94% in 2003 to 3.50% in 2010. In addition, this was the early stage of rapid urbanization growth in Chengdu (average annual growth rate of the urban population of 1.10%; average annual growth rate of the built-up area of 10.02%, Figure 6). The coordinated development of urban construction and eco-

environmental protection has not been afforded sufficient attention. Consequently, there has been a significant decrease in the UGS within both Chengdu's original and expanding urban areas, accompanied by a notable decline in the quality of these areas (Table 3).

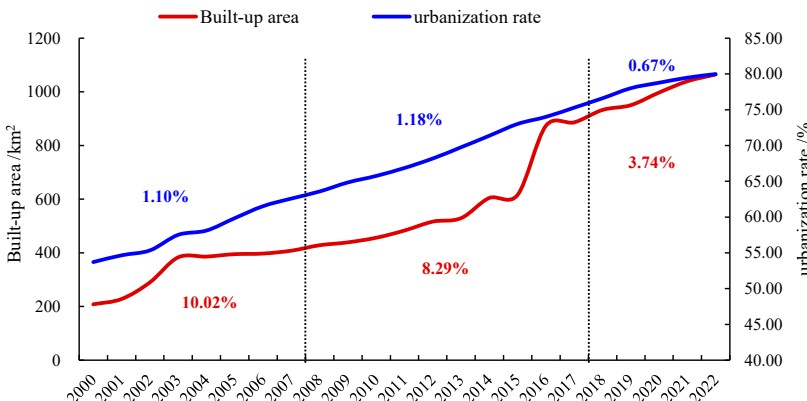

**Figure 6.** Changes in built-up area and urbanization rates in Chengdu (2000–2022).

In the ecological garden city construction stage, with the promulgation of a large number of policies and planning documents, the construction standards and implementation measures of UGS in Chengdu improved, resulting in a significant improvement in the UGS quality in the urban center (Table 3). For example, in the document *Opinions of Chengdu Municipal People's Government on the Construction of Ecological City*, the construction goals of significantly improving the eco-environmental quality and enhancing the function of urban–rural ecosystems were put forward. Another example is the new version of *Urban Green Space System Planning in Chengdu*. That document proposed that the urban center should be planned to form a green space system structure of "one district, two rings, nine corridors, seven rivers, and multiple parks"[3]. However, Chengdu was still in a stage of rapid population and land urbanization (Figure 6) and the greening policy did not significantly improve the situation in that the UGS area decreased significantly in both the original and expanding urban areas (Table 3).

During the national park city construction stage, there was a significant slowdown in both population and land urbanization rates (Figure 6). The result was a corresponding deceleration in the reduction in the UGS area compared to the previous two stages (Table 3). At the same time, the improved rate in UGS quality was also significantly lower than that of the ecological park city construction stage (Table 3). This may be attributed to the fact that the document *Overall Program for Chengdu to Build a Park City Demonstration Area Implementing the New Development Idea* was formally approved in 2022 and the specific measures for the national park city construction had not yet been put into practice during the years 2018–2022. On the other hand, the promulgation of the 2018 document *Work Program for Increasing Quantity and Improving Quality of Central City Landscaping and Greening* resulted in the improvement of the UGS quality in original urban areas to an even greater degree than that in the expanding urban areas (Table 3).

In 2017, Chengdu formally proposed the comprehensive objective of urban development as "advancing eastward, expanding southward, controlling westward, reforming northward, and central excellence"[4]. The conceptual prototype of this plan can be traced back to the document *Chengdu City Master Plan (1995–2020)*, which was released at the end of the 20th century. Combined with the findings on the changes in UGS quantity and quality in different regions, the greening rate of expanding urban areas in the eastern and southern regions declined significantly more than that of the original urban areas in the first two stages of the greening policy (Figure 4a). Moreover, the expanding urban areas in the southern and eastern regions showed a small decrease in UGS quality during the ecological garden city and the national park city construction stages, during which the quality of green space generally improved (Figure 4b). This confirms the urban develop-

ment goals of "advancing eastward" and "expanding southward". The northern region was the earliest urbanization expansion area and the urban construction was dominated by the transformation of old facilities ("reforming northward") into the 21st century. Therefore, the UGS declined to a relatively low extent in the northern region; the decline in the original and expanding urban areas was similar, with the highest improvement in the UGS quality (Figure 4a,b). However, the western region of the urban development control area ("controlling westward") had a degree of decline in UGS that was similar to the eastward advancement and southward expansion areas (Figure 4a). That is, from the perspective of changes in UGS quantity, the implementation of the "controlling westward" urban development goal has not been satisfactory thus far.

## 4. Discussion

### 4.1. Comparative Analysis

According to the *Chengdu Statistical Yearbook*, by the end of 2022, the greening rate of Chengdu's built-up area was 44%. The results of this study also show that the greening rate of the Chengdu city center was about 44.61% in 2022 (Table 3) so this finding is basically consistent with the official statistics. In addition, one study, which had an accuracy of 91.40%, showed that the green space rate in Chengdu was 37.71% in 2018 [41]. In this study, the greening rate of Chengdu Urban Center was 37.63% in 2018 (Table 3), a value which is, again, basically consistent with previous studies. These findings indirectly verify the credibility of the data and methodology used in this study.

It has been demonstrated that the greening rate in densely populated urban areas of China has experienced a rapid decline since the start of the 21st century, with larger urbanized regions exhibiting more significant losses in UGS. Furthermore, newly developed regions have suffered significantly greater reductions in UGS compared to pre-existing built-up regions [56,57]. During the environmental protection model city (2000–2008) and ecological garden city (2008–2018) construction stages in Chengdu, there was a significant decrease in the greening rate within the urban center, with a greater decline observed in expanding urban areas compared to the original urban areas (Table 3). Changes in the UGS quantity in Chengdu are consistent with the overall changes in UGS experienced during China's urbanization process. In recent years, large greening areas in China's urbanized areas have been concentrated in the suburban areas, which have been integrated into the built-up areas by urban expansion, resulting in an increase in the greening level of the built-up areas [36,58]. In this study, the spatial range of Chengdu Urban Center was designed to undergo dynamic changes. With the original urban area integrated with the expanding urban area with higher greening rates to form the new urban center, the greening rate of the new urban center can be restored or enhanced compared with the old urban center (Table 3, Figure 5a).

A recent study on UGS in 338 prefectural-level cities (including provincial-level cities) in China showed that the NDVI of original and expanding urban areas declined by an average of 0.97% and 2.13% per year from 2000 to 2010, respectively. In addition, the NDVI of the original urban area increased by an average of 0.97% per year while that of the expanding urban area declined by an average of 0.28% per year from 2010 to 2018 [28]. In this study, during the period from 2000 to 2008, the NDVI of original and expanding urban green spaces in Chengdu experienced average annual decreases of 1.51% and 2.70%, respectively. This suggests that the implementation of greening policies during the environmental protection model city construction stage in Chengdu was less effective than the national average in a similar period. From 2008 to 2018, the annual average NDVI of original and expanding urban areas in Chengdu increased by 2.30% and 0.42%, respectively, figures which were significantly higher than the national average during the corresponding period. This indicates that Chengdu vigorously developed the greening business in the ecological garden city construction stage and the improvement of UGS quality came to the forefront of the country.

Economically developed cities in eastern China (e.g., Beijing, Shanghai, Xiamen, etc.) have begun to increase the UGS area in their urban centers [37,50,59], which indicates that a gap still exists between the UGS construction in Chengdu and the eastern region in terms of the quantitative dimension. Although the gap is most likely due to the relatively late urbanization of Chengdu, UGS planning should aim to accelerate the shift from UGS area reduction to growth in Chengdu. However, compared with the capital cities in China's central and western regions, the effectiveness of UGS construction is significantly more remarkable in Chengdu. For example, urban expansion in Xi'an and Wuhan continues to come at the expense of large areas of green space and large differences in greening rates within the urban centers [60,61]. In this study, the results show that the greening rate in Chengdu has decreased by less than 5% since 2018 and that there was little difference in the greening rate and UGS quality between subregions (Table 3, Figure 5). In particular, the effectiveness of UGS construction in Chengdu's urban center is higher than that of any prefecture-level city in the Chengdu–Chongqing Urban Agglomeration, including the other core city, Chongqing [62].

### 4.2. Opportunities and Challenges

In February 2022, the *Overall Program for Chengdu to Build a Park City Demonstration Area Implementing the New Development Idea* was officially approved by the National Development and Reform Commission. This means that the construction of the park city demonstration area in Chengdu has officially entered the landing period. The development goals proposed in the document include: the park city demonstration area construction should be achieved via the obvious meeting of established goals by 2025 and the park city demonstration area construction should be fully completed by 2035. Subsequently, Chengdu has formulated the *Action Plan for Establishing a Park City Demonstration Area in Chengdu to Implement the New Development Concept (2021–2025)*. That document proposes specific measures for the profound integration of parks and urban spaces, the establishment of a multi-tiered urban ecological greening system, and the continuous enhancement of green space within built-up areas. Predictably, under the goal of building a national park city demonstration area, the series of policies and measures introduced can provide a guarantee for the increase in UGS quantity and improved quality in Chengdu over the next decade.

In January 2023, to emphasize the status of the Chengdu Plain as the Tianfu Granary for guaranteeing national food security, the People's Government of Sichuan Province issued the *Action Program for Building a Higher Level of the Tianfu Granary in the New Era*. In this document, the proposed plan is to replant 100,000 mu of farmland in the ecological corridor area around Chengdu. Currently, academia has widely recognized that UGS is an artificial, semi-natural, and natural ecosystem in urban areas dominated by vegetation, including parks, gardens, forests, grasslands, and nature reserves [10,11]. Cultivated land is not part of UGS. Therefore, a major challenge will be to balance the positioning of the National Park City Demonstration Area and Tianfu Granary and to integrate the construction of UGS with the high-quality farmland in the urban development of Chengdu.

### 4.3. Limitations

With the development of remote sensing technology, high spatial resolution land use data have begun to be applied to more refined UGS studies. Among them, the most representative data are the 2 m land use data that are produced based on GF-1 satellite imagery, as well as the 10 m land use data that are produced based on Sentinel-2 satellite imagery [63–65]. The primary goal of this study is to reveal the characteristics of long-time series changes in UGS at different stages of greening policy implementation. Since the land use data produced by GF-1 and Sentinel-2 satellites do not have long-time series characteristics, this study employs the long-time series 30 m land use data to reveal the quantitative characteristics of UGS in Chengdu. The 30 m land use data cannot accurately reflect small, fragmented UGS (such as neighborhood greening and street trees), a limitation that can lead to some bias in the assessment of the UGS quantity in this study.

With the continuous deepening of the UGS field, the evaluation of UGS quality has expanded from the green vegetation itself to a dimension that focuses on the interaction between people and green spaces [66], such as accessibility, connectivity, and availability [67–69]. This study only evaluated the green vegetation quality in the urban center of Chengdu based on the NDVI. In the future, we plan to explore the impacts of changes in the spatial structure and quality of UGS on human well-being, based on a more accurate identification of UGS in Chengdu.

## 5. Conclusions

This study selects Chengdu, the pioneering park city in China, as a typical case area. By combing through the documents related to greening policies, Chengdu's greening policies can be categorized into three stages: the environmental protection model city construction stage (2000–2008), the ecological garden city construction stage (2008–2018), and the national park city construction stage (2018–2022). The aim of this study is to explore the changing patterns of UGS in the urban center of Chengdu at different greening policy stages, from the perspective of quantity and quality, and focus on the correlation effect of green policies on UGS construction in Chengdu. The findings not only can provide a reference for Chengdu planners to optimize the UGS construction by the formulation of policies but also have implications for other cities to achieve urban sustainable development through UGS construction.

This study finds that the implementation of greening policies in Chengdu has not yet changed the phenomenon of UGS occupation during urbanization; however, the greening policies have significantly improved the quality of the surviving UGS. In the national park city construction stage, the UGS quality in both the original and expanding urban areas improved by 25.25% and 4.66%, respectively. In this study, the boundaries of the urban center changed dynamically and the old urban center (original urban area) formed a new urban center by integrating the suburban areas (expanding urban area) with higher greening rates. This led to the greening rate of the urban center returning to the 2000 level (44.21%) in 2022 (44.61%). The spatial difference of UGS change reflects the urban development goal of Chengdu, which is "advancing eastward, expanding southward, and reforming northward". However, the implementation effect of the "westward control" has not been satisfactory.

In the future, UGS construction in Chengdu will face both opportunities and challenges. With the implementation of a series of policies and measures to create a park city in Chengdu, the effect of the UGS's increased quantity and improved quality is worth looking forward to. However, it is noteworthy to emphasize the status of the Chengdu Plain as the Tianfu Granary for guaranteeing national food security. In the process of constructing the national park city demonstration area, integrating the construction of UGS and high-quality farmland may become a challenge for the urban sustainable development of Chengdu. Urban tourism agriculture combines the functions of agricultural production and recreation and can provide positive cultural and emotional value for urban residents. Therefore, breaking through the traditional UGS concept and promoting the organic integration of tourism agriculture and the UGS system may become an innovative path to meet the challenge.

**Author Contributions:** Conceptualization, W.D.; methodology, W.D. and Z.Y.; software, K.L. and W.D.; validation, K.L. and W.D.; formal analysis, K.L. and W.D.; investigation, W.D. and H.Y.; resources, W.D. and Y.M.; data curation, K.L.; writing—original draft preparation, K.L. and W.D.; writing—review and editing, Z.Y., H.Y. and Y.M.; visualization, K.L. and W.D.; supervision, H.Y.; project administration, W.D.; funding acquisition, W.D. and H.Y. All authors have read and agreed to the published version of the manuscript.

**Funding:** This research was funded by the National Natural Science Foundation of China (No. 42130508, No. 42301335) and the Startup Foundation in Sichuan Normal University (No. kyqd20220954).

**Data Availability Statement:** The primary data used in this paper are openly available in the data sources. The activity data are from the National Ecosystem Science Data Center (http://www.nesdc.org.cn/sdo/detail?id=60f68d757e28174f0e7d8d49 accessed on 31 January 2024), Peng Cheng Laboratory (https://pan.baidu.com/s/1dt8SILPrLCJZBr1NlskrOA?pwd=7tgm and https://data-starcloud.pcl.ac.cn/zh/resource/14 accessed on 31 January 2024), and Earth System Science Data (https://zenodo.org/records/8176941 accessed on 31 January 2024).

**Conflicts of Interest:** Author Yutong Mu was employed by the company Xi'an Yaozhizhongchuang Land Survey and Planning Co., Ltd. The remaining authors declare that the research was conducted in the absence of any commercial or financial relationships that could be construed as a potential conflict of interest.

## Notes

1. Data source: https://www.kearney.com/industry/public-sector/global-cities/2022 (accessed on 31 January 2024)
2. Data source: https://data.cnki.net/yearBook/single?id=N2023070129 (accessed on 31 January 2024)
3. Notes: "One district" refers to the Ring City Ecological Zone and "two rings" pertain to both the Jinjiang Ring City Park and the 50-meter-wide green belt situated on either side of the Third Ring Road. The term "nine corridors" refers to the nine green traffic corridors formed by the main traffic arteries radiating outward from the center of the city while the term "seven rivers" refers to the green ecological corridors formed by seven tributary waterways (such as the Jinjiang River) within the highway around the city. Finally, "multiple parks" are manifested in various forms, such as comprehensive parks, specialized parks, country parks, and community parks.
4. Notes: Priority is given to ecology and low-to-medium development in the eastern region. A new urban form is being created through expansion in the southern region while better quality and more sustainable developments are achieved through control in the western region. Finally, the quality of regional cities is transformed in the northern region, industrial structure is upgraded, and the industrial capacity level is optimized in the central region.

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
