# Peer review of "Spatio-Temporal Pattern of Urban Green Space in Chengdu Urban Center under Rapid Urbanization: From the Policy-Oriented Perspective"

_land, doi:10.3390/land13040443_

Round 1

Reviewer 1 Report

Comments and Suggestions for Authors

Thank you for sharing your findings regarding the spatio-temporal patterns of Urban Green Space (UGS) in Chengdu's central urban area over a period of rapid urbanization. The study aims to evaluate how UGS has changed over time, both quantitatively and qualitatively, under different stages of greening policies from 2000 to 2022. Upon reviewing the manuscript, some issues have come to my attention, and I will elaborate on them below:

Abstract

The authors need to provide more details on the research methodology employed, including data collection methods, and analysis techniques.

Introduction

The introduction needs major improvements.  

The introduction lacks clarity regarding the specific gaps in previous research that this study aims to address. It's essential to clearly articulate which aspects of understanding regarding the relationship between urbanization, greening policies, and UGS remain unexplored or insufficiently addressed by existing literature.

It is not clear what is the main contribution of the current study.

The theoretical framework or conceptual basis guiding the study's approach and analysis is not clearly outlined.

What does it mean by Urban Greening Space in this study? The authors need to clearly conceptualise and operationalise the key terms used in the study.

The authors need to elaborate further on the specific impact of different greening policies implemented during the studied period and how they influenced UGS trends.

The introduction should explicitly link the study's research objectives to the broader context of urban sustainability and policy-guided construction of UGS.

Results

Compare UGS patterns and policy effectiveness in Chengdu with other cities undergoing similar rapid urbanization processes, offering broader insights.

Discussion

The discussion section begins with a comparative analysis of the greening rate in Chengdu's built-up area, aligning with official statistics. While this comparison is informative, the authors should provide more context regarding how these findings contribute to the broader understanding of urban green space trends in China. It would be beneficial to elucidate why certain areas experienced greater declines in UGS and how these trends relate to broader urbanization patterns.

Throughout the discussion, the authors should consistently link their findings back to the research objectives outlined in the introduction. This would help reinforce the significance of their findings and provide a clear narrative thread for readers to follow.

Comments on the Quality of English Language

Minor editing of English language required.

Author Response

Dear reviewer:

Thank you for your comments on our manuscript entitled “Spatio-temporal Pattern of Urban Green Space in Chengdu Urban Centre under Rapid Urbanization: From the Policy-Oriented Perspective” (Land-2883661). Those comments are valuable and very helpful for revising and improving our paper. We have revised the manuscript carefully according to the comments. Please find our detailed responses below to all these comments/suggestions in the attachment (the replies are highlighted in red in this file and revised manuscript). Thank you again for everything you have contributed and look forward to your approval.

Kind regards,  

Wenpeng Du

Reviewer 2 Report

Comments and Suggestions for Authors

It is an interesting work, and the manuscript is well organized and innovative. However, some issues still should be addressed. Thus, the authors should consider these different points:

1.       It would be better to have a Graphical Abstract. A well-drawn graphical abstract promotes readers' understanding.

2.       The author's introduction needs to be optimized, and we suggest that the author evaluate what needs to be improved in the introduction according to the following criteria.

·       What is the problem to be solved?

·       Are there any existing solutions?

·       Which is the best?

·       What is the main limitation of the best and existing approaches?

·       What do you hope to change or propose to make it better?

·       How is the paper structured?

3.       The map of the study area should indicate latitude and longitude.

4.       The authors are advised to add maps of land cover and land use.

5.       Figure 4 & 5.Changes in the quantity and quality of UGS…” Percentages should not be quantities, , as the total amount changes, the same percentage of land area will also change.

6.       Several new publications in this domain could not be ignored, which should be added in the revised form such as: https://doi.org/10.1177/21582440231208851; https://doi.org/10.3390/land11050652; https://doi.org/10.3390/atmos12121625.

7.       The conclusions need to be revised.

·       Summarize your findings and conclusions from the perspective of the full text and give your opinion on the question (i.e., how your findings answer your research question)

·       Restate the most important findings

·       State the implications of your work (meaning)

·       Make recommendations for future work (inferences)

Comments on the Quality of English Language

The English of your manuscript should be improved before resubmission. We strongly suggest that you obtain assistance from a colleague who is well-versed in English or whose native language is English.

Author Response

(The authors gave the same response as above.)

Reviewer 3 Report

Comments and Suggestions for Authors

In my opinion, there are two major concerns with this paper

  1. Issues with Data Sources: 1. The data sources are not clearly described and it is not clear where most of the data, especially the vegetation index data, were obtained; 2. The data are in discrete years, which is difficult to be representative because there is a high degree of volatility; 3. Why were these discrete years chosen? What is representative or why?
  2. As an argument for policy implementation, this paper lacks a scientifically rigorous method of derivation and seems to me to be a comparison of vegetation indices combined with the appropriate stages. Such a result clearly lacks a scientific basis.
  3. As compensation, the paper could also compare the results with neighboring areas that have not undergone policy implementation, is it the policy that is causing this? Or is it caused by contextual factors such as climate change? This paper lacks sufficient driver analysis.
  4.  

Author Response

(The authors gave the same response as above.)

Reviewer 4 Report

Comments and Suggestions for Authors

Comments and Suggestions for Authors:

The topic of the study is interesting and actual. Generally, the presentation of the study divided into proposed stages is correct, however there are some weaknesses related to the presentation of selected sections/subsections.

1. The title is clear. Key words are well selected and in line with the topic. 

2. The Abstract is well organized and includes information about the study.

3. The Introduction includes main aspects related to conducted study, also the relation to references (other studies) is well presented.

The aim of the study is clear (lines: 95-98).

4. Section 2. “Materials and Methods” has good order of information, however, most subsections need some development. Subsection 2.2. “Selection of Vegetation Index” – the indexes (NIRv, NDVI, EVI) used for the study are just listed, while they need some more description/explanation what they include to make them more clear for readers/scientists. The same is applied to data sources listed in section 2.3. “Data Source and Preprocessing” subsection 2.3.1 “Data Source” (CLCD, GUB, GAIA) - this elements need improvement, what is also related to the presentation of methods.  Subsection 2.4. “Methods” – the description is very general, even superficial in my opinion. The aspects selected by other authors in used method require a broader explanation than just references to publication, etc. (line 155). Generally, the whole section needs improvement.

5. Section 3. “Results and Analysis” should be rather called “Results” while the element of analyses is a part of results. The presentation of results is well organized, includes main and important information of studied aspects (qualitative and quantitative relations) and is supported by comments, also graphs and figures. The subsection 3.3. “Relevance to greening policies” is valuable.

6. Discussion is rather synthetic and insufficient in its present form. Most parts are related rather to general information of China’s UGS than details from conducted study on Chengdu area. The results must be more deeply discussed, also in relation to similar/different results from other studies – more relation to other literature items is needed to increase the value of this study. It would be also valuable to find out if identified trends are unique or similar problems currently occur in other areas, not necessarily in China - this approach would increase the scientific soundness of the study.

Subsection 4.3. “Shortcomings and Prospect” – it would be better to call the shortcomings directly as "limitations", then present them clearly, and indicate how they can be solved in further studies, etc.

8. Conclusions – they need some development to be more explicit, currently they are not very deep. Also some recommendations could be included to increase the role of conducted study, and maybe its practical impact on the improvement of Chengdu planning policy.

Summing up, the presented of material in selected subsections of the manuscript needs some improvement/development. I can’t recommend publication of the manuscript in its present form – it needs revision.

Author Response

(The authors gave the same response as above.)

Reviewer 5 Report

Comments and Suggestions for Authors

Authors present a very interesting research (that fits perfectly the aim of the Journal) on the importance of Urban Green Space (UGS) in the urban environment under the rapid urbanization process of already built areas and the spread of greening policies. Authors select Chengdu as case study underling as the city-region is a clear example to investigate the spatio-temporal characteristics of UGS.

I underline just two minor issues:

1) section 2.1: please add a brief explanation of each phase of green policy promulgation;

2) fugure 2: please add the localization of Chengdu in relation to mainland China.

Author Response

(The authors gave the same response as above.)

Round 2

Reviewer 1 Report

Comments and Suggestions for Authors

Dear authors,

Thank you for sharing the revised manuscript and for taking my comments into consideration. I appreciate the effort you've put into addressing them. Although the authors provided the English Proofreading Certificate, I noticed a grammatical error in line 98. Please double-check the contents for grammar.

Comments on the Quality of English Language

Although the authors provided the English Proofreading Certificate, I noticed a grammatical error in line 98. Please double-check the contents for grammar.

Author Response

Dear Reviewer:

Greetings! Thank you for your contribution to improving the quality of our articles. In this latest revision, we have carefully checked the whole text and revised possible grammatical problems (Red highlighting in the manuscript).

Kind regards,  

Wenpeng Du

Reviewer 3 Report

Comments and Suggestions for Authors

Thanks for the reply, but I don't think this is an effective revision, as the author avoided all the core concerns. 

  1. The study was designed with three stages, but the second stage was not supported by data, and I am still not quite sure why the authors divided it that way.For example, in the first stage, the last policy was set in 2003, but it is divided into 2008.
  2. As I said, the randomness of simply considering a single year is significant, especially with NDVI. This is one of the reasons why I think it is very inappropriate for the study to consider only qualitative research. Indeed, there are a number of current approaches to policy efficiency research, such as the DID model. The authors cannot simply say it is difficult and leave it at that, then the work itself can be left alone. Because even the authors themselves can't be sure if it's a function of policy.
  3. The urban-rural comparison is a relatively simple alternative way of analyzing the results, which the authors continue to dismiss. I don't recognize the authors' argument that NDVI variations in urban areas are primarily derived from anthropogenic influences, and in particular not much from meteorological influences. This is clearly contradictory to the results of many research.

Author Response

Dear reviewer

We are very grateful to you for providing comments on the article. In the future work, we will continue to develop methodologies and case studies of UGS change mechanisms under the greening policies.

Yours sincerely,

Wenpeng DU

Reviewer 4 Report

Comments and Suggestions for Authors

I appreciate all works done by the Authors, most suggestions has been introduced.

The presentation of methods has been developed and improved making them more clear for readers. Presentation of results and discussion has been also improved eliminating some former generalizations. Authors added some more relations to other studies / literature / what is valuable. Conclusions are not much developed, but a bit better organized.

Summing up, the general presentation of all stages of conducted study has been improved, thus the manuscript can be published in its present form in my opinion.

Author Response

Dear Reviewer:

Greetings! Thank you for your contribution to improving the quality of our articles.

Kind regards,  

Wenpeng Du